# A Simple Indoor Localization Methodology for Fast Building Classification Models Based on Fingerprints

**David Sánchez-Rodríguez** [1,2,*,†] , **Itziar Alonso-González** [1] [,2,†], **Carlos Ley-Bosch** [1,2,†]
**and Miguel A. Quintana-Suárez** [2,†]

1   Institute for Technological Development and Innovation in Communications,
    University of Las Palmas de Gran Canaria, Campus Universitario de Tafira,
    35017 Las Palmas de Gran Canaria, Spain; itziar.alonso@ulpgc.es (I.A.-G.); carlos.ley@ulpgc.es (C.L.-B.)
2   Telematic Engineering Department, University of Las Palmas de Gran Canaria,
    Campus Universitario de Tafira, 35017 Las Palmas de Gran Canaria, Spain; mangel.quintana@ulpgc.es
*   Correspondence: david.sanchez@ulpgc.es; Tel.: +34-928-458-047; Fax: +34-928-451-380
†   These authors contributed equally to this work.

**Abstract:** Indoor localization has received tremendous attention in the last two decades due to location-aware services being highly demanded. Wireless networks have been suggested to solve this problem in many research works, and efficient algorithms have been developed with precise location and high accuracy. Nevertheless, those approaches often have high computational and high energy consumption. Hence, in temporary environments, such as emergency situations, where a fast deployment of an indoor localization system is required, those methods are not appropriate. In this manuscript, a methodology for fast building of an indoor localization system is proposed. For that purpose, a reduction of the data dimensionality is achieved by applying data fusion and feature transformation, which allow us to reduce the computational cost of the classifier training phase. In order to validate the methodology, three different datasets were used: two of them are public datasets based mainly on Received Signal Strength (RSS) from different Wi-Fi access point, and the third is a set of RSS values gathered from the LED lamps in a Visible Light Communication (VLC) network. The simulation results show that the proposed methodology considerably amends the overall computational performance and provides an acceptable location estimation error.

**Keywords:** indoor localization; methodology; fingerprint; classification; feature fusion; principal components

## 1. Introduction

Over the last two decades, localization systems have become a subject of great interest due to the need to provide services to users according to their locations. In fact, more and more applications are being developed every day involving user location awareness [1]. Therefore, precise positioning remains a crucial requirement, which is not properly covered in many localization-based systems, such as applications for monitoring people with disabilities or robotic systems.

Thus, precise indoor localization is still a critical missing component, which has gained an increasing interest in a wide range of location-based applications, such as robotics, tracking disabled people, etc.

In outdoor environments, the Global Positioning System (GPS) is the most popular navigation system based on satellites. However, GPS-driven navigation does not work well where there is no line-of-sight with GPS satellites, as happens in inside buildings [2,3]. Furthermore, a 3D localization cannot be determined with GPS; only longitude and latitude information are available. Therefore,

systems capable of sensing location in indoor environments have to be designed to provide services that fulfill the aforementioned requirements. In the literature, many indoor localization-based solutions using other signals, such as Wi-Fi [4], Bluetooth [5], ZigBee [6], RFID [7], or ultrasound [8], have been proposed. Among all of them, Wi-Fi networks have had great attention paid to them, mainly because of their low cost, mature standardization state, and wide deployment.

On the other hand, indoor localization systems based on fingerprints have become a promising approach thanks to their ability to determine user position by means of received signal patterns, such as a collection of RSS values obtained from different access points, in which case there is no need for additional hardware to gather RSS values [9]. Most research works based on fingerprinting methods use machine learning algorithms to learn the relationships among positions and RSS values.

Much work has been published about indoor localization using Wi-Fi infrastructure and fingerprinting, with the aim of reducing system complexity and improving accuracy. In [10], Principal Components (PC) were obtained by processing RSS measurements for the purpose of simplifying and reducing the radio map data handling. Compared to traditional approaches, their method reduced mean error by 33.75%, and complexity was decreased by 40%. The authors in [11] proposed the use of a method of data analysis based on Kernel Principal Component Analysis (KPCA) to remove radio map data redundancy, combined with an algorithm for classifying and grouping reference points on the basis of the Affinity Propagation Clustering (APC) method. Then, they employed Maximum Likelihood (ML) estimation for positioning, obtaining a location accuracy by a margin of 3 m in up to 94% of the cases, having an improvement of 38% compared to the use of ML-based estimation alone.

In [12], they employed relative RSS values and a K-Nearest Neighbor (KNN) algorithm based on Spearman distance to enhance the accurateness of localization under conditions of multipath signal attenuation and environmental changes. Their results showed deviations of up to 2.7 m for 80% of the evaluated samples when having a shadow fading factor of 5 dB. In [13], the authors proposed a technique based on relative neighbor RSS values for radio map building and a Markov chain prediction model for localization. They obtained a stable accuracy in conditions of environmental changes and device heterogeneity, with results of a 1.5-m average error. The authors in [14] analyzed the effects of beacon nodes' placement on the obtained results by several clustering methods employed to reduce positioning time. By evaluating the positioning time and the positioning error as performance metrics, they proposed an optimum beacon node layout scheme, which ensured coverage visibility within the location area, thus improving the accuracy in all tested clustering methods.

In [15], a crowdsourcing localization system was proposed, in which each user could contribute to the construction of the radio map. To mitigate the variation of RSS values due to the use of different devices, a linear regression algorithm combined with a graph-based learning method was applied, to correlate RSS values at nearby locations. According to their results, localization accuracy was improved with an average error of 1.98 m. In [16], the authors proposed a hybrid system combining Wi-Fi with Bluetooth Low Energy to determine user localization, with coarse and fine accuracy, respectively. They employed the KNN algorithm to determine user positions, with accuracy up to 1.47 m and accuracy up to 1.81 m for 90% of the time.

A major weakness of fingerprinting techniques based on RSS is that accuracy is easily affected by the spatial and temporal variation due to the multipath effect. Thus, Channel State Information (CSI) has recently attracted research efforts in indoor localization because CSI is a fine-grained value from the PHY layer, which describes the amplitude and phase on each subcarrier of the OFDM systems, yielding sub-meter-level accuracy [17]. For instance, the authors in [18] used network features obtained from the amplitude and phase information of individual subcarriers and a visibility graph method to determine the location using machine learning algorithms. In [19], the authors used a deep network with four hidden layers to explore the features of wireless channel data and obtain the optimal weights as fingerprints. Although the average error of CSI-based systems is less than RSS-based systems, the former requires more computation time in the training phase to analyze fine-grained features.

Substantial progress has been accomplished in the development of solutions for localization in indoor environments, with the achievement of highly-precise systems. However, a major drawback of those systems is that they have both high demands on computation time and energy consumption. In many situations, elapsed time to build the system and energy constraints are the most important parameters that restrict the fast deployment of localization-based services where an acceptable accuracy can be allowed, and no high accuracy is required. Temporary environments, such as emergency situations due to natural catastrophes, could be candidates to implement localization systems with these restrictions. In addition, the computational complexity of indoor localization systems must be kept in mind when developing for portable devices, in which restrictions usually apply not only in computation capacity but also in energy consumption.

Thus, in this manuscript, a methodology for fast building of an indoor localization system is proposed. Unlike other approaches, few training samples are required to build the dataset, resulting in the system cost of gathering data being cut. This is due to the combination of feature fusion and feature transformation, allowing one to retain valuable information. In addition, both feature fusion and feature transformation are computationally light. Hence, the amount of features can be reduced, with the consequent reduction in the computational load required to estimate the location. Such dimensionality reduction can be a useful technique for processing high-dimensional datasets, such as in high-density Wi-Fi environments, while still retaining as much of the variance in the dataset as possible. Thus, the performance of the classifier can be further enhanced when the discarded information is redundant noise [10]. Three different datasets were used to validate the proposed methodology. Two of them are public datasets based mainly on RSS from different Wi-Fi access points (from the University of Mannheim and the University of Yuan Ze), and the third is an RSS dataset built using a simulation tool where the direct component and multipath reflections of the optical signal from LED lamps are included.

The paper is organized as follows: Section 2 describes the phases of the proposed methodology. Section 3 explains the datasets used for validating the methodology. Next, in Section 4, the experimental results are discussed, and the performance and robustness of our methodology are analyzed. Finally, in Section 5, the conclusions are presented.

## 2. Methodology

The approach followed in this research work consisted of a methodology formed by four phases: data gathering or dataset building, feature fusion, feature transformation, and classification. The methodology is presented in Figure 1. First, a dataset is built with valuable information for each reference location in the environment, such as RSS from Wi-Fi access points, orientation from a compass, Bluetooth beacons, or landmark information. Next, these features are fused into a vector to gather the key properties in every reference location. After that, this vector is transformed into principal components, reducing the data dimensionality and improving the computational performance of the system. Finally, these principal components are applied as input parameters in a classification-based machine learning algorithm to build an indoor localization model. In the following sections, each phase is explained.

### 2.1. Fingerprint Dataset

In order to build a fingerprint dataset, a task is addressed to collect the valuable information or features to characterize the indoor environment, such as RSS values of all detected beacon frames from different access points, device orientation measurements, the distance to landmarks at each reference location, etc. Hence, each reference location is a known position and is represented by its own fingerprint. All the valuable collected data constitute the fingerprint of the environment and compose the dataset. Thus, a dataset can be represented by $\Psi$, as is shown in Equation (1), where $\varphi_{i,j}^{\theta}[\tau]$ is the RSS measurement collected at the $i^{\text{th}}$ reference location and $j$ represents the access point from which RSS was gathered. $\theta$ indicates additional sensors' information. For instance, if a compass is

available, $\theta$ denotes the orientation angle of the device. Furthermore, the number of reference locations is represented by $R$, the total amount of access points by $A$, and the index of RSS samples by $\tau = 1, \dots, N$, where at each reference location, $N$ RSS samples are collected.

$$\Psi = \begin{pmatrix} \varphi_{1,1}^{\theta}[\tau] & \varphi_{1,2}^{\theta}[\tau] & \cdots & \varphi_{1,A}^{\theta}[\tau] \\ \varphi_{2,1}^{\theta}[\tau] & \varphi_{2,2}^{\theta}[\tau] & \cdots & \varphi_{2,A}^{\theta}[\tau] \\ \vdots & \vdots & \ddots & \vdots \\ \varphi_{R,1}^{\theta}[\tau] & \varphi_{R,2}^{\theta}[\tau] & \cdots & \varphi_{R,A}^{\theta}[\tau] \end{pmatrix} \tag{1}$$

On the other hand, multipath effects such as diffraction, reflection, and scattering cause unstable RSS values over time, being a major drawback of indoor localization systems based on fingerprints. Thus, at a particular location, the RSS oscillates close to the average value [20]. Hence, in order to reduce the effects of this RSS unstableness, an arduous task of data collection has to be carried out to build a huge dataset. However, as is demonstrated, in Section 4, the proposed methodology is robust with regard to the training dataset size, and therefore, the efforts and the number of collected samples to build the dataset can be less.

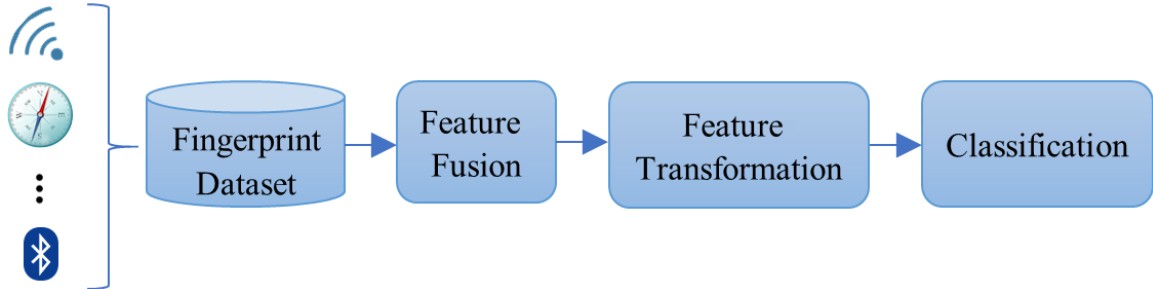

**Figure 1.** The proposed methodology for fast building classification models of indoor localization.

### 2.2. Feature Fusion

All valuable collected information for each reference location is fused, concatenating the features as in Equation (2), in which each row represents an instance for the classification algorithm. Hence, each row contains mainly RSS values, $\varphi_i$ from different access points, and additional information from other sensors, such as orientation information $\theta$ and the reference location $RL_j$.

$$\begin{bmatrix} \varphi_1 & \varphi_2 & \cdots & \varphi_A & \theta & RL_1 \\ \varphi_1 & \varphi_2 & \cdots & \varphi_A & \theta & RL_1 \\ \vdots & \vdots & \ddots & \vdots & \vdots & \vdots \\ \varphi_1 & \varphi_2 & \cdots & \varphi_A & \theta & RL_R \end{bmatrix} \tag{2}$$

### 2.3. Feature Transformation

Once the fusion phase has been carried out, feature transformation is carried out by reducing the number of variables while still retaining much of the information in the original dataset. This transformation is assessed using Principal Components Analysis (PCA) algorithm [21], which is probably the best-known and most widely-used dimension-reducing technique. The correlation of the valuable information of the environment is used to compute the principal components, which replace the features stored in the dataset. In fact, in order to cut down a huge dataset of features to a small dataset that still includes most of the information in the large dataset, the PCA algorithm is used. Hence, the number of columns of previous feature fusion, Equation (2), is reduced. Thus, the number of operations carried out to generate the model by the classifier is diminished, and therefore, the computation time is also decreased.

*2.4. Classification*

In this phase, three machine learning algorithms were assessed to validate the proposed methodology. The algorithms used for classification are the following: K-Nearest Neighbor (KNN), AdaBoost, and Support Vector Machine (SVM). Next, a brief description of these algorithms is outlined.

2.4.1. K-Nearest Neighbor

KNN is an algorithm employed in machine learning solutions that relies on the prediction of new data classification made on the basis of matching the closest samples existing in the feature space [22]. KNN makes the decision of selecting the distances to the K-nearest point to the data under observation for predicting the class that is more alike. Thus, class prediction is estimated according to the mere majority of neighbors. To use KNN, the IBk method defined in [23] was employed in this work.

2.4.2. AdaBoost

The boosting technique allows one to enhance the accuracy of classifiers formed by decision trees. This technique is based on combining the predictions of multiple base or weak classifiers in order to obtain a more powerful classifier. One of the most popular algorithms used for boosting in classification is AdaBoost [24]. This algorithm is based on both iterative and adaptive methods, combining base models of the same kind so that the base models obtained from preceding iterations have an influence on the performance of each new model. The experiments carried out in this work used the C4.5 algorithm [25] as the weak classifier to obtain a decision tree as the base model. In this manuscript, the AdaBoostM1 and J48 implementations of AdaBoost and C4.5 algorithms were used, respectively.

2.4.3. Support Vector Machine

SVM [26] is a robust supervised machine learning algorithm applied for classification, with the goal of creating non-overlapping partitions by mapping the data as parts of a higher-dimensional space. By classifying geometric parameters and from the training data, SVM obtains the optimal hyperplane that divides the data into two fully-differentiated classes. Linear, polynomial, and non-linear kernels were tested for separating the classes. In this paper, an LIBSVM library [27] was used, which implements both the C-Support Vector Classification (C-SVC) and nu-Support Vector Classification (nu-SVC) methods.

**3. Datasets**

Three different datasets have been used to validate the proposed methodology in this manuscript. Two of them have been built taking measures in real environment and contain RSS values gathered from several Wi-Fi access points and orientation information from a compass. The third dataset was obtained using a software tool used to simulate VLC networks and contains RSS values obtained from several simulated LED lamps.

*3.1. Mannheim University Dataset*

The dataset was built at the University of Mannheim, Germany, concretely on the second floor of an office building [28]. This dataset can be accessed at CRAWDAD, a Community Resource for Archiving Wireless Data supported by Dartmouth College. It contains a set of RSS values gathered with a Lucent Orinoco PCMCIA network card from IEEE 802.11 base stations, and the receiver orientation angle was collected with a C8051F350 Digital Compass.

Figure 2 shows the layout area of the testbed environment, which is nearly 36 m long and 15 m wide. There were four Lancom L-54g and five Linksys WRT54GS access points present. Six of them were positioned in the testing area, which was marked by squares in the plan. The rest of the base stations were placed on other floors of the building.

The gray dots depict the 166 reference locations, which were distributed with a spacing of 1 m. In the offline phase, eight orientations (with intervals of 45°) were used to measure the RSS values at each reference location. For each orientation and location, 110 RSS values were consecutively taken every 250 ms. Hence, the dataset had 146,080 instances, which each instance containing nine RSS values and one orientation value. Furthermore, the black dots represent the 60 locations used for the online phase where each gathered 110 RSS values and one orientation angle. Thus, the online phase contained 6600 instances.

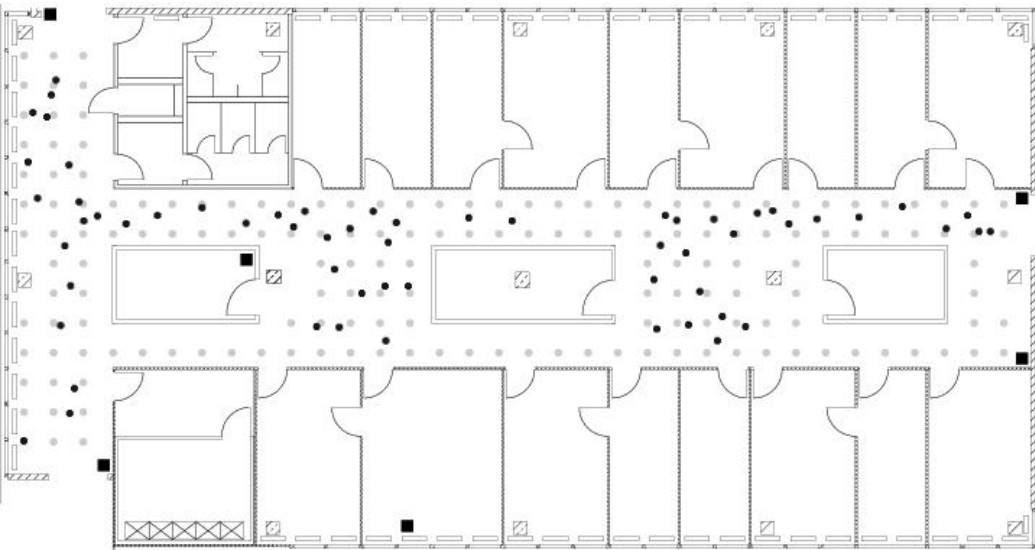

**Figure 2.** Test environment at the University of Mannheim.

### 3.2. Yuan Ze University Dataset

The dataset was built at the University of Yuan Ze, Taiwan, concretely on the fourth floor of the telecommunications building [29]. Figure 3 shows the layout area of the testbed environment, which is 36.5 m wide and 63.5 long, and there were 41 access points present. A total of 70 reference locations with a spacing of 1.2–2 m was defined to gather RSS values. In addition, four orientations (with intervals of 90°) were set to measure the RSS values at each reference location. For each orientation and location, 50 RSS values were gathered. Hence, the dataset had 14,000 instances, each instance containing 41 RSS values and one orientation value. Furthermore, 23 different locations were used for the online phase, recording also 50 RSS values and four orientation angles for each one. Thus, the online phase contained 4600 instances.

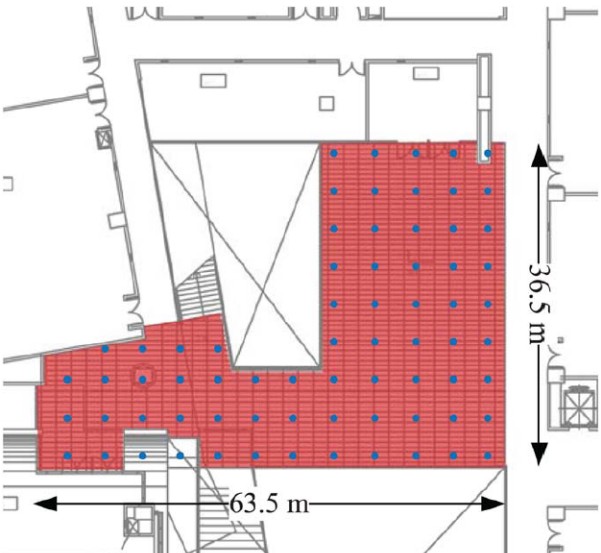

**Figure 3.** Test environment at the University of Yuan-Ze.

### 3.3. VLC Dataset

For this dataset, we have used the Communication and Lighting Emulation Software (CandLES) [30], which allowed us to calculate RSS values and other features, such as the channel impulse response, $h(t)$, in a VLC network. This software implements a fast multi-receiver channel estimation for indoor optical wireless communications [31]. The calculations for this model take into account, among others, factors such as localization of the light sources and receivers, the reflectance of the obstacles and walls, the optical sensor area in each receiver, and also the number of signal reflections. The scenario designed to build the dataset is the room shown in Figure 4, which is 4 m wide, 4 m long, and 3 m in height without obstacles. As light sources, 16 LED lamps (red triangles) of 15 W were set up, distributed in a $4 \times 4$ grid with 1-m separation from each other. As receivers, 361 devices (blue circles) were set up in a $19 \times 19$ grid placed 20 cm equidistant. All receivers had the following configuration: a photo sensor of 100 mm$^2$ with an optical concentrator gain of 10, a value of 60° for the Field Of View (FOV), and a concentrator refractive index of 2.73. Furthermore, in order to consider a 3D localization, three different heights of receivers were evaluated. The distances from the floor were 125, 100, and 75 cm. Lastly, a different reflectivity index was set for the ceiling, wall, and floor, that is 0.69, 0.58, and 0.09, respectively.

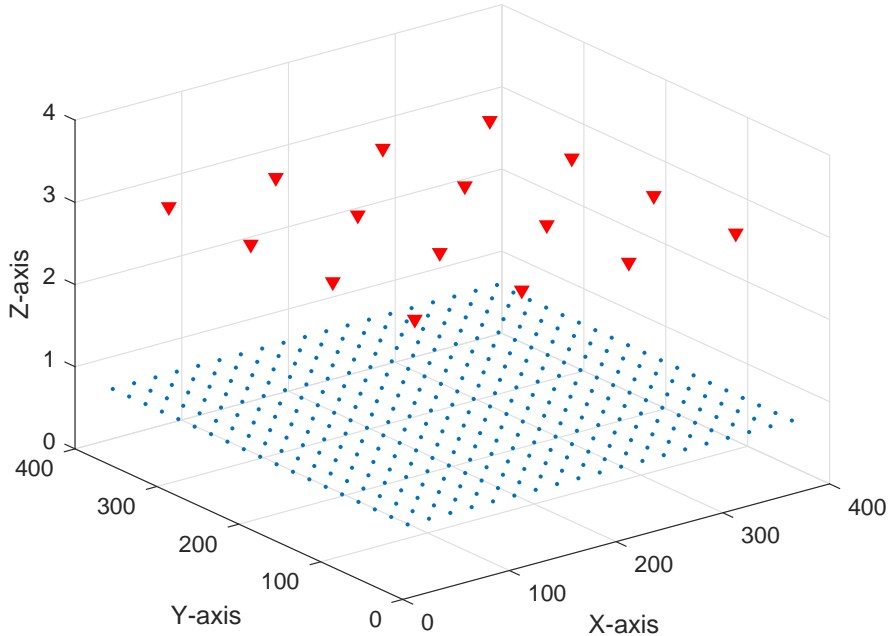

**Figure 4.** VLC simulation scenario at 0.75 m from the floor.

Next, a description of the performed simulations is described:

- Line of sight: A total of 11 simulations was implemented considering only the RSS of the optical signal in the line of sight, that is only the direct component was taken into account. In the first simulation, all receivers pointed out to the ceiling with an angle of 90 degrees with the floor. For the following simulations, every receiver was set with a different orientation angle chosen between 75° and 105°.
- Two paths: Both the direct component and the first reflection were considered at each receiver. Eleven simulations were done as in the line of sight case.
- Three paths: Both the direct component and two reflections were considered at each receiver. Eleven simulations were done as in the line of sight case.
- Four paths: Both the direct component and three reflections were considered at each receiver. Eleven simulations were done as in the line of sight case.

Therefore, a total of 44 simulations was performed on each receiver. This led to 47,652 (361 receivers × 3 layers × 44) instances for the dataset, which each instance containing 16 RSS values, one from each LED lamp. Therefore, the full dataset was formed by 762,432 RSS values.

### 3.4. Summary of Datasets

A summary of the main attributes of the dataset used to validate the proposed methodology in this manuscript is shown in Table 1.

**Table 1.** Summary of the datasets.

| Dataset | Instances | Locations | RSS Measurements for Each Instance | Instances per Location and Orientation | Orientations | 3D |
|---|---|---|---|---|---|---|
| Mannheim | 146,080 | 166 | 9 | 110 | 8 (each 45°) | No |
| Yuan Ze | 14,000 | 70 | 41 | 50 | 4 (each 90°) | No |
| VLC | 47,652 | 361 | 16 | 44 | No | Yes |

## 4. Experimental Results and Discussion

In this section, the performance results of the proposed methodology in the three aforementioned datasets are described and discussed. Experiments were focused on comparing accuracy, error distance, and computation time using several machine learning algorithms based on classification. Furthermore, the robustness of the methodology was evaluated using different training dataset sizes. The accuracy was computed as the number of instances correctly classified divided by the total number of instances used for testing, Equation (3). The expected distance from the misclassified instance (estimated location) and the real position were used as the error measure in our system. This error was obtained by calculating the Euclidean distance between these points, and the arithmetic mean was computed from the results of the experiments.

$$\text{Accuracy}(\%) = \frac{\text{Instances Correctly Classified}}{\text{Total Number of Test Instances}} \times 100. \tag{3}$$

In order to validate the experimental results and to ensure statistical independence, all experiments were repeated 100 times, averaging the results, and 10-fold cross-validation was used. The methodology was implemented using the Weka API [32]. All experiments were carried out on an Intel Core i7 3.4 GHz/32 GB RAM non-dedicated Windows machine.

### 4.1. Feature Transformation

In order to transform the data fusion into principal components, PCA algorithm was used in conjunction with a Ranker search. Dimensionality reduction was accomplished by choosing enough eigenvectors to account for 95% of the variance in the original data. Table 2 shows the features transformation. As can be observed, for the Yuan Ze and VLC datasets, the number of features was considerably reduced, and therefore, the number of computations carried out for the classifiers was also greatly reduced. However, most of the features of the Mannheim dataset were considered outstanding, and for that reason, the number of principal components was almost equal to the number of features. This may be due to the fact that all access points have been strategically placed in the environment, and therefore, all of them provide relevant information.

**Table 2.** Feature transformation results.

| Dataset | Features | Principal Components |
| --- | --- | --- |
| Mannheim | 9 RSS + 1 Orientation | 9 |
| Yuan Ze | 41 RSS + 1 Orientation | 23 |
| VLC | 16 RSS | 6 |

### 4.2. Setting of Classifiers

In order to find the best setting of each classifier several experiments were carried out varying the key parameter of classifiers with the three datasets. Both fusion and transformation of features were applied to the dataset before being used as input to the classifier.

For the KNN algorithm, experiments were implemented to find the optimal k parameter, varying from k = 1 to k = 5. For the AdaBoost algorithm, confidenceFactor, c, was varied from 0.20 to 0.40 in steps of 0.05. Lastly, for the SVM algorithm, the C-SVC and nu-SVC implementations were used, and three kinds of kernels were used to find the best accuracy: linear, polynomial, and Radial Basis Function (RBF). For both polynomial and RBF kernels, the gamma parameter was varied from $2^{-10}$–$2^4$.

Tables 3 and 4 show accuracy results obtained with different values of the key parameter for KNN and AdaBoost, respectively. Table 5 shows the best accuracy for each kernel using SVM as the classifier. The gamma value for the best accuracy is specified in brackets.

**Table 3.** Accuracy for different values of the k parameter (KNN).

| Dataset | KNN | | | | |
|---|---|---|---|---|---|
| | k = 1 | k = 2 | k = 3 | k = 4 | k = 5 |
| Mannheim | **60.89%** | 55.79% | 55.96% | 55.37% | 54.73% |
| Yuan Ze | **70.72%** | 69.95% | 69.65% | 69.29% | 69.05% |
| VLC | **91.01%** | 84.31% | 82.56% | 78.39% | 75.83% |

**Table 4.** Accuracy for different values of the c parameter (AdaBoost).

| Dataset | AdaBoost | | | | |
|---|---|---|---|---|---|
| | c = 0.2 | c = 0.25 | c = 0.3 | c = 0.35 | c = 0.4 |
| Mannheim | 51.02% | **51.24%** | 51.13% | 51.21% | 51.22% |
| Yuan Ze | 57.18% | **57.20%** | 57.20% | 57.20% | 57.19% |
| VLC | 86.72% | 86.76% | 86.79% | 86.77% | **86.80%** |

**Table 5.** Accuracy for different kinds of kernels and SVM. C-SVC, C-Support Vector Classification.

| Dataset | C-SVC | | | nu-SVC | | |
|---|---|---|---|---|---|---|
| | Linear | Polynomial | Radial | Linear | Polynomial | Radial |
| Mannheim | 9.3% | **36.20%** (4) | 28.77% (4) | 7.29% | 32.25% (0.25) | 33.39% (4) |
| Yuan Ze | 67.30% | **70.67%** (0.125) | 70.38% (0.125) | 55.59% | 59.08% (0.125) | 63.64% (0.125) |
| VLC | 75.69% | **95.64%** (1) | 88.74% (16) | 81.94% | 78.22% (0.125) | 89.20% (16) |

As can be observed, for the KNN algorithm, the best results were obtained with k = 1 for all datasets. For the AdaBoost algorithm, a confidenceFactor of 0.25 yielded the best accuracy with the Mannheim and Yuan Ze dataset, and for the VLC dataset, the best accuracy was obtained when the confidenceFactor was set to 0.4. However, regardless of the value of the confidenceFactor, all settings had similar accuracy. Lastly, for the SVM algorithm, the best results were reached using the C-SVC machine and the polynomial kernel for all datasets. Furthermore, this classifier provided the best accuracy results, except for the Mannheim dataset, although they were similar to the results obtained when KNN was used. The rest of the experiments described in this manuscript were implemented using the key parameter value that yielded the best accuracy.

*4.3. Evaluation of the Methodology*

Table 6 shows the performance results obtained from the proposed methodology taking into account the accuracy, error distance, and the elapsed training time by the classifier to build the model. Table 7 shows the performance results when the dataset was directly used as input to the classifier, that is without data fusion and transformation of features. As can be seen, with regard to the accuracy and error distance, similar results were yielded in both experiments. Even so, in most of the experiments for the Yuan Ze and VLC datasets, the results obtained using the proposed methodology were slightly worse because of the fact that features transformation removed some useful information. For the Mannheim dataset, principal components removed redundant information, and hence, the proposed methodology improved the accuracy by about 15% using the KNN algorithm.

**Table 6.** Performance of the proposed methodology.

| Dataset | KNN | | | AdaBoost | | | SVM | | |
|---|---|---|---|---|---|---|---|---|---|
| | Accuracy | Error Distance | Training Time | Accuracy | Error Distance | Training Time | Accuracy | Error Distance | Training Time |
| Mannheim | 60.89% | 1.46 ± 2.47 | 46.2 ms | 51.24% | 1.92 m ± 2.83 | 993 s | 36.20% | 2.08 ± 2.43 | 812 s |
| Yuan Ze | 70.72% | 0.94 ± 1.77 | 31.7 ms | 57.20% | 1.37 ± 1.99 | 16.1 s | 70.27% | 1.04 ± 1.88 | 5.9 s |
| VLC | 91.01% | 1.9 cm ± 0.06 | 37.6 ms | 86.80% | 3.1 cm ± 0.08 | 325 s | 95.64% | 1.0 cm ± 0.04 | 17.7 s |

**Table 7.** Performance without data fusion, nor the transformation of features.

| Dataset | KNN | | | AdaBoost | | | SVM | | |
|---|---|---|---|---|---|---|---|---|---|
| | Accuracy | Error Distance | Training Time | Accuracy | Error Distance | Training Time | Accuracy | Error Distance | Training Time |
| Mannheim | 45.98% | 1.35 m ± 2.4 | 58.5 ms | 51.40% | 1.89 ± 2.75 | 1351 s | 30.39% | 2.35 ± 2.53 | 886 s |
| Yuan Ze | 70.27% | 1.03 m ± 1.89 | 45.3 ms | 63.81% | 1.23 m ± 1.29 | 24.6 s | 70.25% | 1.08 m ± 1.95 | 6.3 s |
| VLC | 92.36% | 1.6 cm ± 0.05 | 70.1 ms | 87.12% | 2.7 cm ± 0.07 | 659 s | 97.86% | 0.4 cm ± 0.03 | 29.1 s |

With regard to the elapsed time to build the model, it can be appreciated that this time was considerably less when the proposed methodology was used. For the Yuan Ze and VLC datasets, the computation time to train the classifier was less. Even so, for VLC dataset, the time reduction was about 50% for all classifiers. This was due to the fact that the number of features was considerably reduced in both datasets when the transformation into principal components was carried out (see Table 2), and hence, the number of computations was greatly reduced. The same applies to the Yuan Ze dataset. However, for the Mannheim datasets, the number of principal components was almost equal to the number of original features, and therefore, the number of computations was similar. Thus, the elapsed time to build the model was similar in both experiments.

On the other hand, regardless of the dataset, the KNN algorithm was the classifier that yielded the best results, both in accuracy and computation time.

Lastly, Figures 5–7 show the Cumulative Distribution Function (CDF) using the proposed methodology. As can be seen, for the Mannheim dataset, the error was about 2 m with a 75% probability using KNN as the classifier, and furthermore, it was about 5 m with a 90% probability using any classifier. Something similar occurred for the Yuan Ze dataset: the error was about 2 m with a 80% probability using KNN as the classifier, and the error was about 4 m with a 90% probability using any classifier. For both datasets, at the 90th percentile and above, the accuracy was comparable in all classifiers. For the VLC dataset, most of the test instances were correctly classified, and most of the misclassified instances were about 20 cm for all classifiers, that is the error was about 20 cm with a 95% probability. Therefore, the experimental results demonstrated that the proposed methodology yielded an acceptable accuracy, with a reduction of computation time.

*4.4. Methodology Robustness*

In order to validate the robustness of the proposed methodology, the efficiency of this approach was tested by varying the training dataset size from 20–80%. All experiments were performed using the fastest classifier, that is the classifier implemented by the KNN algorithm.

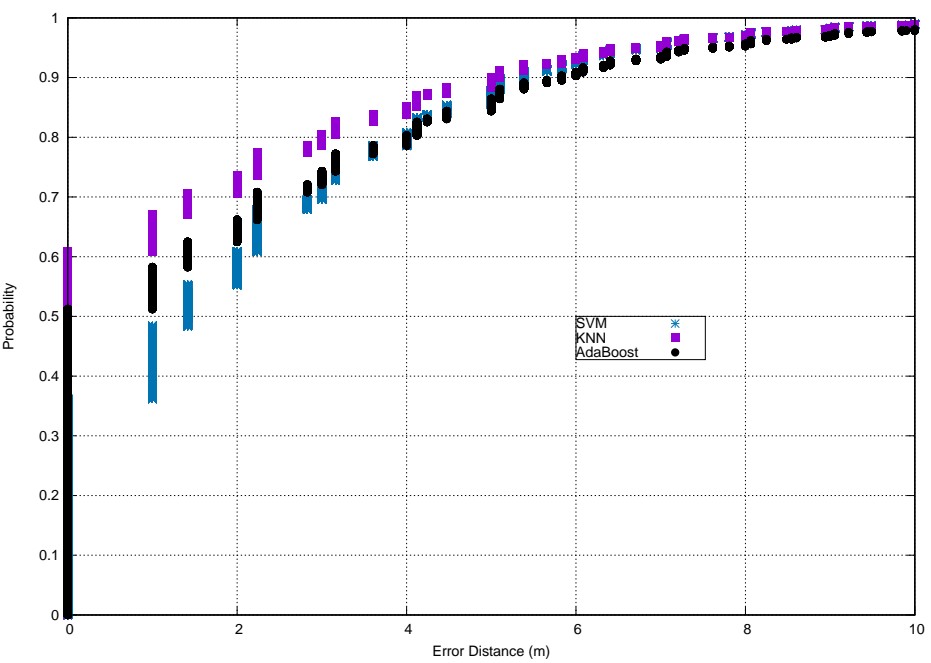

**Figure 5.** CDF for the Mannheim dataset.

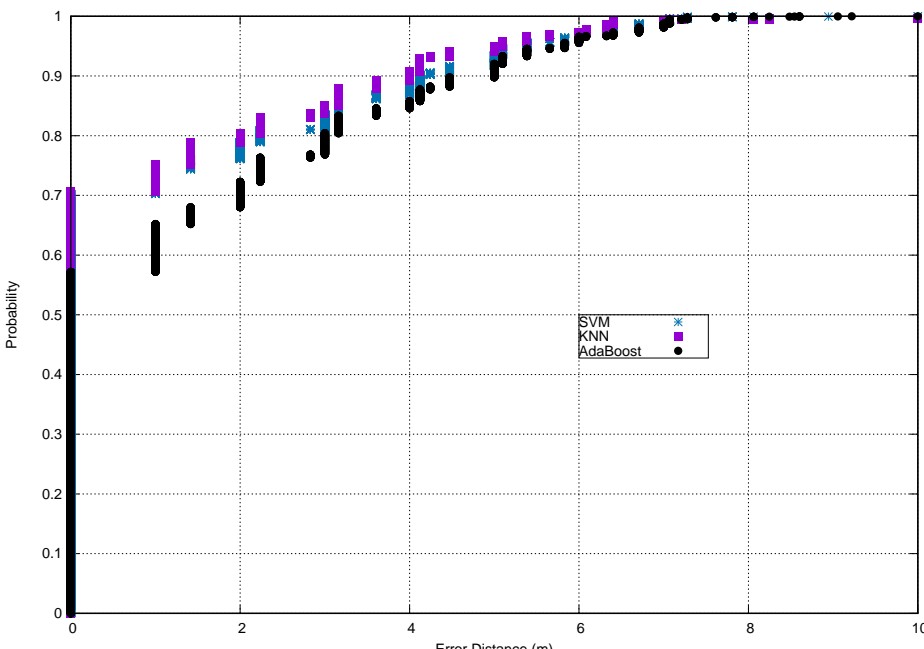

**Figure 6.** CDF for the Yuan Ze dataset.

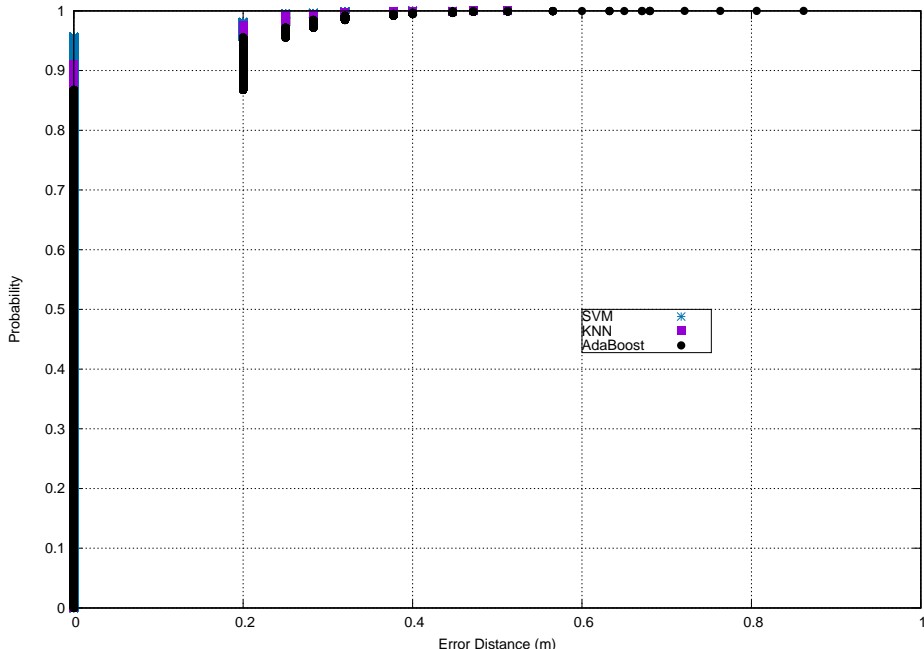

**Figure 7.** CDF for the VLC dataset.

Table 8 shows the experimental results obtained by varying the training size. As can be seen, the system accuracy increased when the training size did, yielding good results with a low number of training samples. Thus, for the Yuan Ze dataset, the accuracy only decreased about 7% when the size of the training dataset was 20% of the total. For the Mannheim and VLC datasets, the methodology can be considered equally robust using 60% of the training samples, with only about a 5% accuracy reduction. For other training sizes, this approach decreased in effectiveness. Furthermore, precision, recall, and F-measure measurements also followed a similar behavior related to the accuracy. Thus, the measurements increased with the training size, keeping the values close to 0.76 when using only a 40% training size for the Yuan Ze and VLC datasets. This shows that the methodology was effective for the fast building of an indoor localization system, for which the reduction in the size of the training set and the reduction of the data dimensionality offered a time savings; although, it may reduce the performance in terms of error distance.

**Table 8.** Performance results varying the training size and using the KNN classifier.

| Dataset | Training Size | Accuracy | Average Error | Training Time | Precision | Recall | F-Measure |
|---------|--------------|----------|---------------|---------------|-----------|--------|-----------|
| Mannheim | 20% | 42.77% | 2.25 m ± 2.84 | 32.6 ms | 0.430 | 0.428 | 0.428 |
| | 40% | 51.39% | 1.82 ± 2.68 | 37.2 ms | 0.515 | 0.514 | 0.514 |
| | 60% | 56.19% | 1.63 m ± 2.58 | 39.0 ms | 0.562 | 0.562 | 0.562 |
| | 80% | 59.35% | 1.52 m ± 2.51 | 42.1 ms | 0.594 | 0.594 | 0.593 |
| Yuan Ze | 20% | 63.12% | 1.21 m ± 1.95 | 21.9 ms | 0.715 | 0.631 | 0.655 |
| | 40% | 67.44% | 1.09 m ± 1.92 | 23.4 ms | 0.761 | 0.674 | 0.699 |
| | 60% | 69.33% | 1.03 m ± 1.85 | 28.8 ms | 0.818 | 0.693 | 0.728 |
| | 80% | 69.46% | 1.02 cm ± 1.86 | 31.1 ms | 0.805 | 0.695 | 0.732 |
| VLC | 20% | 59.85% | 8.8 cm ± 0.11 | 27.1 ms | 0.602 | 0.599 | 0.599 |
| | 40% | 75.75% | 5.2 cm ± 0.09 | 29.2 ms | 0.760 | 0.758 | 0.758 |
| | 60% | 84.23% | 3.4 cm ± 0.08 | 33.1 ms | 0.845 | 0.842 | 0.842 |
| | 80% | 88.90% | 2.4 cm ± 0.06 | 34.8 ms | 0.891 | 0.889 | 0.889 |

Finally, Table 9 shows a comparative analysis with other state-of-the-art indoor localization methods. As can be seen, the results obtained in this work were similar to other research works, being able to achieve a low error distance, and our even research outperformed some recent works.

**Table 9.** Comparative study in terms of accuracy.

| References | Positioning Algorithm | Data | Error Distance |
|---|---|---|---|
| [11] | KPCA + Clustering + Maximum Likelihood | RSS from APs | 1.76 m |
| [12] | KNN based on Spearman distance | RSS from APs | 3.25 m |
| [10] | PCA + Probabilistic Method | RSS from APs | 2.05 m |
| [19] | Deep network | CSI | 1.83 m |
| [33] | Fusion + Probabilistic Distribution | RSS from APs + Orientation | 1.65 m |
| [34] | Genetic algorithm | RSS from LED lamps | 2–5 cm |
| [35] | FDM and TDOA | Time and phase differences of packets | 0.020 mm |
| This work | PCA + KNN | RSS from LED lamps | 1.9 cm |
| | Fusion + PCA + KNN | RSS from APs + Orientation | 1.46 m |

## 5. Conclusions

Indoor localization is a subject that is still open due to the growing interest in location-based services. Nevertheless, sometimes, there are scenarios in which the speed of deployment of a localization system is more important than its accuracy. Thus, in this manuscript, a methodology for the fast building of an indoor localization system has been proposed. Data fusion and feature transformation were used to reduce the computation time.

In order to validate the methodology, three different datasets were used. Two of the them were datasets from the University of Mannheim and the University of Yuan Ze. These datasets contained mainly RSS from different Wi-Fi access points. The third set of RSS data was obtained by simulation where both reflections and the direct component of the optical signal of a VLC network were considered in the simulations. The results presented in this manuscript and obtained experimentally have proven that the proposed methodology provides an acceptable accuracy for all datasets, with the KNN algorithm being the fastest classifier. Furthermore, devices with computational power and energy consumption constraints profit from this methodology due to its low computational complexity. Lastly, the robustness of the methodology was tested using small training datasets. As might be expected, the effectiveness of the system was less when we used a smaller training dataset. However, even reducing the training dataset by 60%, the results in terms of accuracy indicated only a reduction around 5% compared to the use of the complete dataset.

**Author Contributions:** Conceptualization, D.S.-R.; investigation, D.S.-R. and I.A.-G.; methodology, D.S.-R.; validation, D.S.-R., I.A.-G., C.L.-B., and M.A.Q.-S.; writing, review and editing, D.S.-R., I.A.-G., C.L.-B., and M.A.Q.-S.

**Funding:** This work has been partially funded by the Consejería de Economía, Industria, Comercio y Conocimiento del Gobierno de Canarias (CEI2018-16), Spain.

**Conflicts of Interest:** The authors declare no conflict of interest.

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
