# Peer review of "A Simple Indoor Localization Methodology for Fast Building Classification Models Based on Fingerprints"

_electronics, doi:10.3390/electronics8010103_

Round 1
Reviewer 1 Report
The manuscript deals with system for indoor positioning based on fingerprinting approach. Authors proposed methodology for fast building classification models. However, description of the proposed methodology does not provide much details and it is not clear how is the proposed approach novel in comparison to other approaches that use features fusion in fingerprinting positioning.
The system was tested on three different datasets, which is nice. Unfortunately, presentation of the results is not good enough. In Tables 3 and 4 accuracy is presented in %, it is not clear how this was calculated. In description of there results "precision" is addressed, however, there is no definition of precision in the manuscript.
Figures 5, 6 and 7 are not cumulative distribution functions, moreover, it is not clear what "CDF" on y-axis stands for.
In table 8, impact of training size on performance of KNN classifier is evaluated. metrics like "Precision", "Recall" and "F-Measure" are used. However, it is not clear how these metrics can be applied to the area of positioning, where position cannot be directly linked to a single classifier. This is due to the fact that measurements should be taken on different positions in training and evaluation phases.
Author Response
Please find the response attached.

Reviewer 2 Report
This paper uses machine learning for indoor localization with three different data sets. First, the paper provides the method for indoor localization including fingerprint dataset, features fusion, features transformation, and classification. Moreover, three machine learning methods are used to validate the proposed method, which are KNN, AdaBoost, and SVM. Finally, the simulation results verify the effectiveness of the proposed method. In order to improve this paper, the following issues should be addressed.
For related work, the authors should discuss other indoor localization methods such as deep learning based on CSI.
For the proposed method, the author should discuss the difference between this method and PCA based indoor localization [18].
For classification, why do the authors choose these three machine learning methods rather than others? What's the strong points for these methods?
For experimental part, the authors should compare the proposed method with other existing methods.
Author Response
Please find the response attached.

Reviewer 3 Report
The authors present a paper about an area that is widely studied and several contributions have been made already. Authors claim that in emergency situations, there is the need to have an indoor tracking more accurate.
The introduction includes a literature review and what I am missing, considering so many studies exist, is a clear definition on the distinction of this work, regarding how it is really different in fast building (as WiFi usage as several constraints on this) and can help in emergency situations and be a benefit regarding other approaches.
The methodology and datasets sections are well described and with enough detail.
Experiment section is also adequate.
As a big number of similar studies are presented in literature, a more applied research would be desirable, showing how in practice this approach could be of benefit in emergency situations, as authors claim to be their motivation in the beginning of the chapter.
Author Response
Please find the response attached.

Round 2
Reviewer 1 Report
"ensure the performance of accuracy" is somehow strange formulation.
"This shows that the methodology is effective for fast building of an indoor localization system which the reduction in the size of training set and the reduction of data dimensionality offer a time saving although it declines in effectiveness."
What is considered by "effectiveness", from my point of view if the system is using less memory and is faster, then it should be more effective.
"The accuracy is computed as the number of instances correctly classified divided by the total number of instances used for testing"
Does that mean all measurements used in the testing were done on exactly same spots as reference measurements for radio map? I would suggest to use different metric to evaluate performance of localization system, since in real world conditions most of the localization requests will come from positions different than reference points.
Author Response
Dear reviewer, thank you for your time in reviewing this manuscript. Please kindly find the answers in the attached file.

Reviewer 2 Report
The authors have addressed all issues. The paper can be accepted.
Author Response
Dear reviewer, thank you for your time in reviewing this manuscript.